# Healthcare utilization in Canadian children and young adults with asthma during the COVID-19 pandemic

Teresa To[1,2,3]*, Kimball Zhang[2,3], Emilie Terebessy[2], Jingqin Zhu[2,3], Christopher Licskai[4,5]

1 Dalla Lana School of Public Health, University of Toronto, Toronto, Ontario, Canada, 2 The Hospital for Sick Children, Toronto, Ontario, Canada, 3 ICES, Toronto, Ontario, Canada, 4 Schulich School of Medicine & Dentistry, Western University, London, Ontario, Canada, 5 London Health Sciences, Victoria Hospital, London, Ontario, Canada

* teresa.to@sickkids.ca

**Data Availability Statement:** We are not able to provide a minimal data set for this study due to privacy, legal, prescribed entity designations, and ethical restrictions. All data used in this study are

## Abstract

Literature is limited regarding the COVID-19 pandemic's impact on health services use in younger Canadian populations with asthma. We utilized health administrative databases from January 2019–December 2021 for a population-based cross-sectional study to identify Ontario residents 0–25 years old with physician-diagnosed asthma and calculate rates of healthcare use. Multivariable negative binomial regression analysis was used to adjust for confounders. We included 716,690 children and young adults ≤25 years. There was a sharp increase of ICS and SABA prescription rates at the start of the pandemic (March 2020) of 61.7% and 54.6%, respectively. Monthly virtual physician visit rates increased from zero to 0.23 per 100 asthma population during the pandemic. After adjusting for potential confounders, rate ratios (RR) with 95% confidence intervals (CI) showed that the pandemic was associated with significant decrease in hospital admissions (RR = 0.21, 95% CI: 0.18–0.24), emergency department visits (RR = 0.35, 95% CI: 0.34–0.37), and physician visits (RR = 0.61, 95% CI: 0.60–0.61). ICS and SABA prescriptions filled also significantly decreased during the pandemic (RR = 0.58, 95% CI: 0.57–0.60 and RR = 0.47, 95% CI: 0.46–0.48, respectively). This Canadian population-based asthma study demonstrated a dramatic decline in physician and emergency department visits, hospitalizations, and medication prescriptions filled during the COVID-19 pandemic. An extensive evaluation of the factors contributing to an 80% reduction in the risk of hospitalization may inform post-pandemic asthma management.

## Introduction

Early in the pandemic, the Centers for Disease Control and the World Health Organization identified a "theoretical" possibility that youth with asthma infected with COVID-19 could experience asthma exacerbation and serious morbidity due to the combined effects on the respiratory tract [1–3]. People with underlying medical conditions, including asthma, are concerned about the potential effects of the virus on their health. Studies examining patients

securely housed at ICES, Ontario, Canada in coded form and are subject to their privacy, legal, prescribed entity designations, and ethical governance, available at www.ices.on.ca/Data-and-Privacy/Privacy-at-ICES (e-mail: privacy@ices.on.ca). While legal data sharing agreements between ICES and data providers (e.g., healthcare organizations and government) prohibit ICES from making the dataset publicly available, access may be granted to those who meet pre-specified criteria for confidential access, available at http://www.ices.on.ca/DAS (e-mail: das@ices.on.ca). The dataset creation plan is available in the supporting information. This does not alter our adherence to PLOS ONE policies on sharing data and materials.

**Funding:** Dr. TT receives a grant (HLTC3968IT) from the Ontario Ministry of Health (https://www.ontario.ca/page/ministry-health). The funders had no role in study design, data collection and analysis, decision to publish, or preparation of the manuscript.

**Competing interests:** The authors have declared that no competing interests exist.

hospitalized with COVID-19 infection have not found over-representation of patients with co-morbid asthma [4–7]. To date, there is no clear evidence to suggest that children and young adults with asthma are at increased risk for severe COVID-19 infection [4–7].

Limited data are available on the impact of the COVID-19 pandemic on the asthma-related health services use (HSU). In general, the pandemic has caused a large decrease across all preventative and elective health care services globally [8]. This has been accompanied by the rapid uptake of telemedicine, especially in urban areas, but this increase has not replaced the lost volume of in-person medical care. Many factors may have contributed to the decline in HSU during the pandemic including societal factors, like city-wide lockdowns and cancellations of non-urgent or elective procedures and care, and viral avoidance strategies such as social distancing and masking. These pandemic variables may also be important factors impacting asthma symptom control and related HSU.

There is limited literature examining the impact of the COVID-19 pandemic in children and young adults with asthma. While recent literature suggests improved asthma control during the pandemic, it has reported mixed results on whether individuals with asthma have greater susceptibility to COVID-19 [9–12]. Understanding how the pandemic has impacted asthma-related HSU and medication use, can provide insights into medication adherence, asthma symptom and exacerbation control during the pandemic. Therefore, this study aims to use population-based health administrative data to examine the change in the of patterns of access to healthcare services during the COVID-19 pandemic in children and young adults with asthma.

## Methods

### Study design & population

The trend of asthma HSU was investigated using a cross-sectional design. The study population included Ontario residents aged ≤25 years old with prevalent asthma in the period of January 1, 2019 to December 31, 2021. We included this age group of children and young adults due to public drug coverage availability among those ≤25 years in Ontario via the Ontario Health Insurance Plan (OHIP) Plus Ontario Drug Benefit (ODB) program. Asthma diagnosis was determined based on an administrative case definition of ≥1 hospitalization for asthma, or ≥2 outpatient visits for asthma in two consecutive years. This definition has been previously validated in Ontario with a sensitivity of 84% and a specificity of 77% [13]. The study population included individuals who met this definition of asthma diagnosis prior to the time period of this study.

Individuals were excluded if they were ever diagnosed with chronic obstructive pulmonary disease (COPD), congestive heart failure, cystic fibrosis, lung cancers, Crohn's disease, ulcerative colitis, rheumatoid arthritis, or bronchiectasis. Individuals were also excluded if they did not have data on age, an Ontario residence code, or a valid Ontario health card number.

### Data sources

This study used routinely collected health administrative data for Ontario. In Ontario, there is a publicly funded single-payer healthcare system. Health administrative data were linked using unique encoded identifiers at ICES, formerly known as the Institute for Clinical Evaluative Sciences. Data on hospital admissions were captured by the Canadian Institute for Health Information Discharge Abstract Database (CIHI-DAD) while data on emergency department (ED) visits were captured through the National Ambulatory Care Reporting System (NACRS). The OHIP claims database captured outpatient physician office and virtual visits. CIHI-DAD, NACRS and OHIP data were available from January 1, 2019 to December 31, 2021. As was

mentioned earlier, public drug coverage is available in Ontario for children and young adults aged < 25 years through the OHIP Plus ODB program starting April 1, 2019. Asthma medication prescriptions filled, as covered by OHIP Plus, were captured through the ODB database. These prescriptions were identified using the Drug Identification Number. Data were available from April 1, 2019 to December 31, 2021. Data on patient characteristics like age, sex, and residence postal code were captured through the Provincial Registered Persons Database. Date of asthma diagnosis was captured through the Ontario Asthma Surveillance Information System (OASIS; https://lab.research.sickkids.ca/oasis/).

## Exposure & outcome definitions

The primary outcome consists of four types of HSU: 1) hospital admission for asthma, 2) an ED visit for asthma, 3) an ambulatory care physician visit (in-person or virtual) for asthma, and 4) asthma prescription medication including inhaled corticosteroids (ICS), alone or in combination, and short-acting beta 2-agonists (SABA). Asthma ED visits or hospital admissions were identified using ICD-10-CA codes (J45 and J46). The primary exposure was the pandemic period defined as March 2020 to December 2021, while the pre-pandemic period was defined as January 2019 to February 2020.

## Covariates

Potential confounders included in multivariable negative binomial regression analyses included age, sex, location of residence (rural/urban), census-based income quintiles and the ON-Marg ethnic concentration quintiles. _Socioeconomic status_ (SES) was measured by proxy, using the census-based income quintiles and the ethnic concentration domain of the Ontario Marginalization Index (ON-Marg) [14]. ON-Marg provided a measure of marginalization at the population-level based on census information. Based on each participant's residence postal code, they were assigned a score from 1 (least marginalized) to 5 (most marginalized) for each dimension. _Residence_ was considered rural if the individual resided in a community with a population of ≤10,000 people, or urban if the opposite was true. Asthma medications were stratified by any ICS or any SABA use.

## Statistical analysis

Monthly asthma HSU rates were calculated with number of encounters as the numerator and the prevalent asthma population as the denominator. Hospital admission and ED visit rates were expressed as per 10,000 prevalent asthma population ≤25 years old, while physician visit rates and asthma prescribed medication rates were per 100 prevalent asthma individuals ≤25 years old. Monthly rates were also calculated stratified by five age groups (0–4, 5–9, 10–14, 15–19, 20–25). Two consecutive pandemic periods: 2020 (March 2020-Janurary 2021) and 2021 (February-December 2021) were included in the regression analysis to allow for balanced comparative observation windows. Multivariable negative binomial regression models were used to examine the rates of health services and asthma prescription medication use while adjusting for all covariates as outlined above. The adjusted rate ratios (RR) of HSU were described with 95% confidence intervals (CI). All statistical analyses were conducted using SAS Enterprise Guide 9.4 [15] and forest plots were generated using the _forestplot_ package in R statistical computing software version 3.3.3 (https://www.r-project.org/). Ethics approval exemption was obtained from the Hospital for Sick Children Research Ethics Board (Toronto, Ontario, Canada). Individual patient consent was not required as this study involved secondary use of health administrative data that were fully anonymized. This is in compliance to the

Tri-Council Policy Statement: Ethical Conduct for Research Involving Humans (TCPS2) Articles 5.4 and 5.5A (https://ethics.gc.ca/eng/tcps2-eptc2_2018_chapter5-chapitre5.html#d).

## Results

### Population characteristics

Table 1 shows that there were 716,690 children and young adults aged ≤25 years old with prevalent asthma in Ontario in 2019. Of these, 7.1% were pre-schoolers (0–4 years), 16.4% early school aged (5–9), 21.4% early teens (10–14), and 55.1% youths and young adults (15–25). The study cohort consisted of 41.9% females, largely from areas of middle to high income quintiles (63.4%), and the majority (92.5%) resided in urban areas.

### Descriptive trends on asthma health services use

**Asthma hospitalizations.** Fig 1A shows the monthly trend of asthma hospital admission rates from January 2019 to December 2021. The total number of asthma hospitalizations in the pre-pandemic period was 2,438 representing an average monthly rate of 2.4 per 10,000 prevalent asthma population. This number dropped to 830 during the pandemic corresponding to an average monthly rate of 0.5 per 10,000, representing a 78.3% decline during the pandemic. This decline was seen in all age groups. The fall seasons (September to November) in both 2019, 2020 and 2021 remain the months with the highest asthma hospital admissions. Asthma hospital admission rates were highest in the youngest children aged 0–4 years both before and during the pandemic.

**Asthma emergency department (ED) visits.** Fig 1B shows the monthly trend of ED visits for asthma from January 2019 to December 2021. The total number of asthma ED visits in the pre-pandemic period was 16,933, representing an average monthly rate of 16.9 per 10,000 prevalent asthma individuals. This number dropped to 8,864 during the pandemic which corresponds to an average monthly rate of 5.6 per 10,000, representing a 66.7% decline during the

**Table 1. Distribution of characteristics of the study population (N = 716,690).**

| Characteristics | | Number | Percentage |
|---|---|---:|---:|
| Age groups | 0–4 | 51,118 | 7.13 |
| | 5–9 | 117,489 | 16.39 |
| | 10–14 | 153,579 | 21.43 |
| | 15–19 | 180,923 | 25.24 |
| | 20–25 | 213,581 | 29.80 |
| Sex | Female | 299,945 | 41.85 |
| | Male | 416,745 | 58.15 |
| Income quintiles | Q1 (lowest) | 130,983 | 18.34 |
| | Q2 | 130,494 | 18.27 |
| | Q3 | 145,047 | 20.30 |
| | Q4 | 153,941 | 21.55 |
| | Q5 (highest) | 153,897 | 21.54 |
| Ethnicity concentration | Q1 (least) | 81,299 | 11.45 |
| | Q2 | 101,025 | 14.23 |
| | Q3 | 122,990 | 17.33 |
| | Q4 | 159,967 | 22.53 |
| | Q5 (most) | 244,611 | 34.46 |
| Rural residence | | 53,400 | 7.47 |

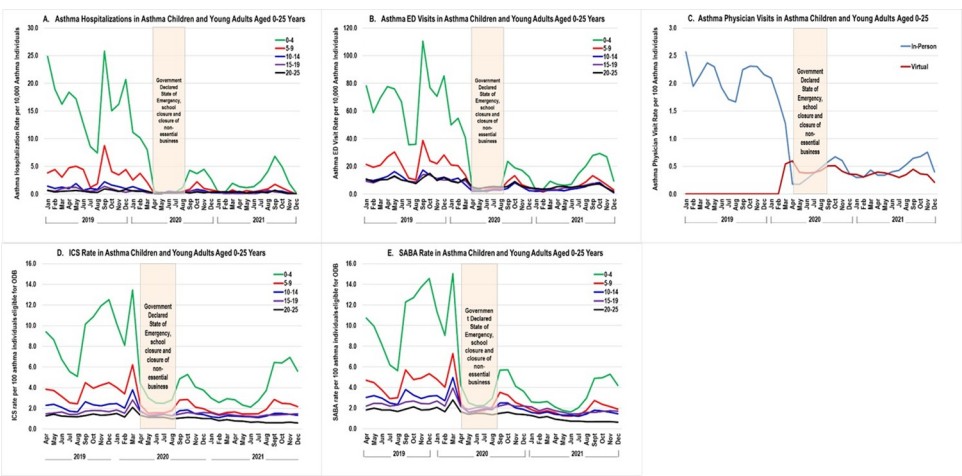

**Fig 1. Asthma health services use rates by age groups, January or April 2019 to December 2021 (N = 716,690).**
Abbreviations: ICS = inhaled corticosteroid; SABA = short-acting beta 2-agonist; ODB = Ontario Drug Benefit.

pandemic. This decline was seen in all age groups. As with asthma hospital admissions, asthma ED visit rates were also highest in the youngest children aged 0–4 years both before and during the pandemic.

**Asthma ambulatory care visits.**   Fig 1C shows the monthly trend of physician in-person and virtual visits for asthma from January 2019 to December 2021. The total number of physician in-person visits in the pre-pandemic period was 214,569, representing an average monthly rate of 2.1 per 100 prevalent asthma individuals. This number dropped to 140,396 during the pandemic which corresponds to an average rate of 0.9 per 100, representing a 58.4% decline during the pandemic. In contrast, the rate of physician virtual visits increased over the 22-month pandemic period. Pre-pandemic, physician virtual visits happened rarely (at a rate close to zero) but sharply increased since March 2020 with a total of 36,301 claims over the 22 months corresponding to an average monthly rate of 0.23 per 100.

**Asthma prescription medication.**   In the pre-pandemic period, the monthly ICS rate was 2.3 per 100 asthma individuals eligible for the provincial drug plan. The ICS rate continued to decrease to an average of 1.53 per 100 during the pandemic, representing a 33.9% decrease. Similarly, the monthly rate of SABA prescription filled decreased from 3.1 per 100 in the pre-pandemic period to 1.8 per 100 during pandemic, representing a 40.4% decrease. In general, the highest asthma medication prescription rates were observed in late summer to fall seasons. At the start of the pandemic, there was a spike in both ICS (61.7% increase) and SABA prescriptions (54.6% increase) filled in March 2020; this spike was observed in all age groups (Fig 1D).

The observed descriptive trends on the rates of HSU and asthma medication prescription filled were similar across sex, SES (income and ethnic concentration quintiles), and urban/rural residence (details are presented in S1–S4 Figs of the Online Supplementary Content).

## Multivariable negative binomial regressions adjusting for potential confounders

**Asthma health services use.**   The multivariable negative binomial regression rate ratios and their corresponding 95% CI of asthma HSU are shown in Table 2 and Fig 2. After adjusting for confounders, the pandemic periods were significantly associated with decreased asthma HSU, specifically in 2021: hospital admissions (RR = 0.21, 95% CI: 0.18–0.24); ED visits

**Table 2. Adjusted rate ratios of asthma health services use based on negative binomial regressions.**

| Covariates | Hospital Admissions | | Emergency Visits | | Physician Visits | |
|---|---|---|---|---|---|---|
| | Rate Ratio | 95% CI | Rate Ratio | 95% CI | Rate Ratio | 95% CI |
| Pre-pandemic (January 2019—February 2020, reference) | 1.00 | | 1.00 | | 1.00 | |
| Pandemic (March 2020—January 2021) | 0.25 | [0.22, 0.28] | 0.37 | [0.35, 0.39] | 0.61 | [0.60, 0.61] |
| Pandemic (February 2021—December 2021) | 0.21 | [0.18, 0.24] | 0.35 | [0.34, 0.37] | 0.61 | [0.60, 0.61] |
| Age 20–25 (reference) | 1.00 | | 1.00 | | 1.00 | |
| Age 0–4 | 19.45 | [16.60, 22.79] | 4.67 | [4.42, 4.94] | 3.38 | [3.33, 3.44] |
| Age 5–9 | 5.31 | [4.51, 6.24] | 1.77 | [1.68, 1.86] | 1.97 | [1.94, 2.00] |
| Age 10–14 | 2.05 | [1.71, 2.45] | 1.00 | [0.94, 1.05] | 1.39 | [1.37, 1.41] |
| Age 15–19 | 1.29 | [1.06, 1.56] | 1.01 | [0.96, 1.06] | 1.04 | [1.03, 1.06] |
| Male (reference) | 1.00 | | 1.00 | | 1.00 | |
| Female | 1.16 | [1.06, 1.27] | 1.22 | [1.18, 1.27] | 1.16 | [1.15, 1.18] |
| Urban (reference) | 1.00 | | 1.00 | | 1.00 | |
| Rural residence | 0.97 | [0.80, 1.18] | 1.82 | [1.71, 1.93] | 0.74 | [0.73, 0.76] |
| Income quintile Q5 (reference) | 1.00 | | 1.00 | | 1.00 | |
| Income quintile Q1 | 2.31 | [1.98, 2.70] | 2.23 | [2.11, 2.37] | 0.93 | [0.92, 0.95] |
| Income quintile Q2 | 1.63 | [1.39, 1.90] | 1.62 | [1.52, 1.71] | 0.98 | [0.97, 1.00] |
| Income quintile Q3 | 1.45 | [1.24, 1.69] | 1.40 | [1.32, 1.48] | 1.00 | [0.99, 1.02] |
| Income quintile Q4 | 1.19 | [1.02, 1.40] | 1.20 | [1.13, 1.27] | 0.99 | [0.98, 1.00] |
| Ethnic concentration Q5 (reference) | 1.00 | | 1.00 | | 1.00 | |
| Ethnic concentration Q1 | 1.74 | [1.47, 2.05] | 2.01 | [1.89, 2.14] | 0.59 | [0.58, 0.61] |
| Ethnic concentration Q2 | 1.77 | [1.53, 2.06] | 1.90 | [1.80, 2.01] | 0.65 | [0.64, 0.66] |
| Ethnic concentration Q3 | 1.56 | [1.35, 1.79] | 1.67 | [1.58, 1.77] | 0.73 | [0.72, 0.74] |
| Ethnic concentration Q4 | 1.32 | [1.15, 1.50] | 1.37 | [1.30, 1.44] | 0.82 | [0.81, 0.83] |

Abbreviations: CI = confidence intervals; Q1 = first quintile; Q2 = second quintile; Q3 = third quintile; Q4 = fourth quintile; Q5 = fifth quintile

(RR = 0.35, 95% CI: 0.34–0.37); physician visits (in-person and virtual combined) (RR = 0.61, 95% CI: 0.60–0.61).

Higher risk ratios for hospital admission or an ED visit were seen in the youngest members of the cohort, females, rural residents, those with lower income, and those who lived in the least diverse communities. This gradient pattern was similar for physician visits.

## Asthma prescription medication

The RR from multivariable negative binomial regression and their corresponding 95% CI of ICS and SABA prescriptions filled are also shown in Table 3 and Fig 2. After adjusting for confounders, the results demonstrated a significant and continuing decreased in ICS and SABA prescriptions filled in the pandemic periods. For example, during pandemic in 2021, the RRs of ICS and SABA prescriptions filled showed a 42% decrease (RR = 0.58, 95% CI: 0.57–0.60) and 53% decrease (RR = 0.47, 95% CI: 0.46–0.48), respectively, compared to the pre-pandemic period.

The RR for ICS and SABA prescription fills were similar across the subgroup analyses. Females were more likely to refill than males, rural residents more than urban, and lower income more than higher income.

## Discussion

This Canadian population-based study followed over 700,000 asthma prevalent children and young adults aged ≤25 years and showed substantial decreases in HSU for asthma during the

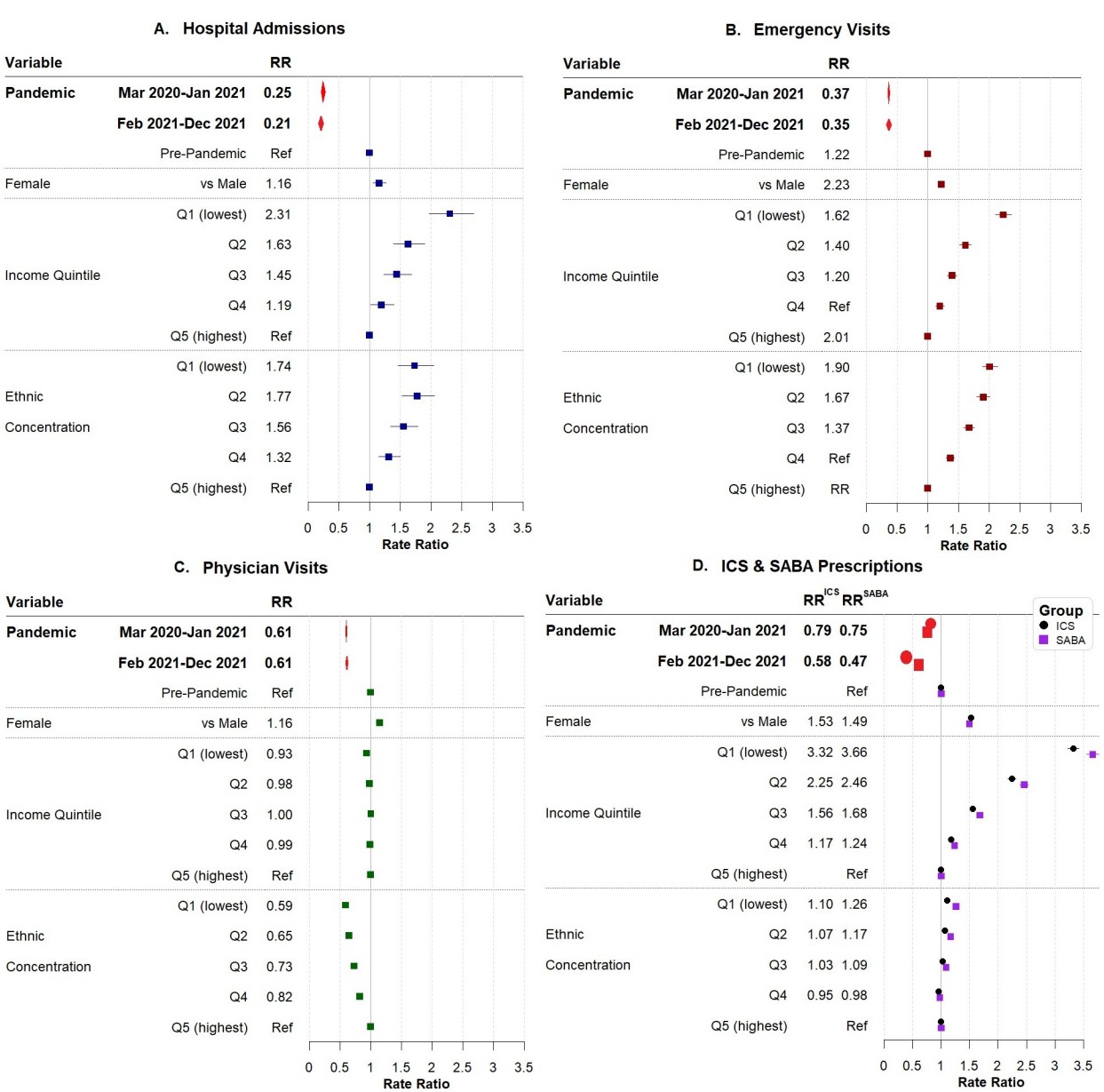

**Fig 2. Forest plot of adjusted rate ratios (RR) of asthma health services use based on negative binomial regressions.** Abbreviations: Ref = reference; RR = rate ratio; Q1 = first quintile; Q2 = second quintile; Q3 = third quintile; Q4 = fourth quintile; Q5 = fifth quintile; ICS = inhaled corticosteroid; SABA = short-acting beta 2-agonist. Note: All regression models were also adjusted for rural or urban residence and age groups.

COVID-19 pandemic. This study distinguishes itself from other COVID-19 pandemic HSU investigations as the first whole health system population-based study and as the most current study with the longest pre-pandemic and pandemic-related observation period (36 months).

The volume of acute healthcare utilization, namely ED visits and hospital admissions for asthma, dropped by over 65% and 75%, respectively, during the pandemic in 2020–2021. Similarly, outpatient healthcare of in-person physician visits decreased by 58%. In contrast, virtual physician visits increased exponentially, although this increase did not offset in-person reductions. The rate of prescribed asthma medications filled, including ICS and SABA, also showed a continuing decrease during the pandemic despite a sharp increase at the start of the

**Table 3. Adjusted rate ratios of asthma prescription medication based on negative binomial regressions.**

| Covariates | ICS Prescriptions | | SABA Prescriptions | |
|---|---|---|---|---|
| | Rate Ratio | 95% CI | Rate Ratio | 95% CI |
| Pre-pandemic (January 2019—February 2020, reference) | 1.00 | | 1.00 | |
| Pandemic (March 2020—January 2021) | 0.79 | [0.77, 0.81] | 0.75 | [0.74, 1.00] |
| Pandemic (February 2021—December 2021) | 0.58 | [0.57, 0.60] | 0.47 | [0.46, 0.48] |
| Age 20–25 (reference) | 1.00 | | 1.00 | |
| Age 0–4 | 22.67 | [21.89, 23.48] | 17.25 | [16.66, 17.86] |
| Age 5–9 | 4.55 | [4.43, 4.68] | 3.86 | [3.76, 3.97] |
| Age 10–14 | 2.31 | [2.25, 2.37] | 2.30 | [2.24, 2.36] |
| Age 15–19 | 1.77 | [1.73, 1.82] | 1.88 | [1.84, 1.93] |
| Male (reference) | 1.00 | | 1.00 | |
| Female | 1.53 | [1.50, 1.56] | 1.49 | [1.47, 1.52] |
| Urban (reference) | 1.00 | | 1.00 | |
| Rural residence | 14.68 | [14.12, 15.27] | 16.35 | [15.75, 16.97] |
| Income quintile Q5 (reference) | 1.00 | | 1.00 | |
| Income quintile Q1 | 3.32 | [3.22, 3.42] | 3.66 | [3.56, 3.77] |
| Income quintile Q2 | 2.25 | [2.18, 2.31] | 2.46 | [2.39, 2.53] |
| Income quintile Q3 | 1.56 | [1.51, 1.61] | 1.68 | [1.63, 1.73] |
| Income quintile Q4 | 1.17 | [1.14, 1.21] | 1.24 | [1.20, 1.27] |
| Ethnic concentration Q5 (reference) | 1.00 | | 1.00 | |
| Ethnic concentration Q1 | 1.10 | [1.07, 1.14] | 1.26 | [1.22, 1.30] |
| Ethnic concentration Q2 | 1.07 | [1.04, 1.10] | 1.17 | [1.13, 1.20] |
| Ethnic concentration Q3 | 1.03 | [1.00, 1.06] | 1.09 | [1.06, 1.12] |
| Ethnic concentration Q4 | 0.95 | [0.93, 0.98] | 0.98 | [0.95, 1.00] |

Abbreviations: CI = confidence intervals; Q1 = first quintile; Q2 = second quintile; Q3 = third quintile; Q4 = fourth quintile; Q5 = fifth quintile; ICS = inhaled corticosteroids; SABA = short-acting beta 2-agonists

pandemic. Of note is the period from April-August 2020 when the Ontario Government declared a State of Emergency with province-wide closures of all schools and non-essential businesses and services. Specifically, during the State of Emergency, hospital admissions and ED visits for asthma in the asthma population "flattened," regardless of age. While there was a general decline in HSU across the population during the pandemic, our adjusted multivariable regression analysis showed that young children (especially those <5 years), females, those who live in rural areas, and those in a low-income quintile or living in the least diverse communities, consistently had the highest asthma HSU.

Our observations in Ontario, Canada is consistent with how pediatric asthma specialists have described their pandemic-related clinical practice modifications globally. Papadopoulos et al. found that COVID-19 significantly impacted pediatric asthma services in an online survey with 91 respondents, from 27 countries across six continents, reporting that 39% ceased physical appointments, 47% stopped accepting new patients, and 75% limited patients' visits [16]. Consultations were almost halved and virtual clinics and helplines were launched at most centers. These survey findings suggest that the outpatient HSU reductions identified in our study are related to planned reductions in health services access related to the pandemic. Papadopoulos surveyed providers early in the pandemic, while our study provided a much longer follow-up period, thereby suggesting that limited outpatient access persisted as the pandemic continued.

The asthma-related outpatient and urgent HSU reductions identified in our study align with those of Moynihan et al., who conducted a systematic review on the impact of the

COVID-19 pandemic on utilization across the spectrum of all healthcare services provided to patients [8]. The review included 81 studies across 20 countries, reporting on >11 million services pre-pandemic and 6.9 million during the pandemic. They found a median 37% reduction in services overall (interquartile range [IQR] −51% to −20%), comprising median reductions for visits of 42% (−53% to −32%) and admissions 28% (−40% to −17%). It is important to differentiate that Moynihan's systematic review relates to all health care services, and is thus not asthma-specific thereby limited to observations during the first four months of the pandemic. Recognizing that asthma is an airways disease and that SARS-CoV-2 is a respiratory virus, there was a need to specifically evaluate HSU specific to an asthma population. The COVID-19 pandemic has prompted an unprecedented expansion of virtual patient care. Our study population showed a significant decline in in-person physician visits which was partially substituted by virtual visits. While many (health care providers and patients) applauded this change [17], others have warned about the possibility that virtual patient care can bring unintended consequences including potentially limiting the patient-provider relationship, quality of examination, the efficiency of health care delivery, missed worsening symptoms and subsequently worse health outcomes in patients [18–21]. Longitudinal studies following-up of patients post-pandemic and post-utilization of virtual care are needed to fully evaluate the benefits and harms/risks of virtual versus in-person health care.

Early in the pandemic, there were concerns that exponentially rising infections from SARS-CoV-2 could lead to an increase in asthma exacerbations and associated rise in ED visits and hospitalizations. The concern was premised on the basis that viral respiratory tract infections are a common cause of asthma exacerbations [22], non-pandemic coronaviruses have been associated with exacerbations [23], and ED visits and hospitalizations increase in children annually on calendar week 38, an increase associated with rising viral infection after returning to school [24]. Fortunately, these initial concerns have not been realized; there has been no identified increase in exacerbations or hospitalization [4–7, 25]. Remarkably, the reverse is observed. This study and others demonstrate a dramatic decrease in urgent health services use including ED visits and hospitalizations. Our study aligns with the systematic review by Yang and colleagues, who determined that asthma exacerbation rates declined during the pandemic (OR = 0.26, 95%CI: 0.14–0.48), along with ED visits (OR = 0.11, 95% CI: 0.04–0.26) but with no significant reduction in hospitalization identified [26]. Guijon and colleagues examined 18,912 pediatric patients and identified significant reductions in exacerbations, ED visits and hospitalizations 78%, 90%, 68%, respectively, compared with pre−COVID-19 2020 [27].

Studies published based on data acquired early in the pandemic describe increased adherence to asthma control medication. Increased adherence may be associated with patient concern that their asthma symptoms could be confused with COVID-19 symptoms [28]. Papadopoulos et al. found that during the pandemic, there was nearly a 2-fold (RR = 1.97, 95% CI: 1.66–2.33) increased adherence to asthma controller medications [16]. In the US, Kaye et al. reported a 14.5% relative increase in daily asthma controller medication adherence from 7,578 patients in January 2020 to March 2020 [28]. Consistent with these studies, our study also found that the rate of prescribed asthma control medications (ICS) filled spiked early in the pandemic. While we also observed a similar spike involving SABA, there are limited literature quantifying asthma rescue medication use early in the COVID-19 pandemic. Instead, most literature on asthma rescue medication like SABA involved longer COVID-19 pandemic periods with little comparisons on usage between the early and later parts of the pandemic [29–31]. These authors found an overall decreased use of SABA during COVID-19.

In a similar vein, our data suggests that asthma control improved during the pandemic. The measures that support that assertion included reduced fills of SABA for relief of asthma symptoms and reduced severe exacerbations measured by ED visits and hospitalization. Of interest,

these control measures improved despite significantly fewer outpatient asthma visits and lower ICS prescription fill rates. Improved control has been described by other authors. In Northeast Italy, Ferraro et al. compared the use of asthma medications in 92 children with asthma, in March-April 2019 versus March-April 2020 and found that more children had their therapy modified during the COVID-19 pandemic lockdown (38% versus 15.2%, p<0.001) and there was a higher level of asthma symptom control [32]. A study of 46,900 children in the United States between March 2020-September 2021 by Rao et al. did not find evidence of an association of COVID infection in children with asthma after controlling for use of ICS. A potential explanation of this "apparent" improved asthma symptom control or a lower COVID infection risk, may be attributable to reduced exposure to potential asthma triggers and respiratory viruses due to lockdown (e.g. traffic-related pollution) [10].

There are several limitations to this study. First, we were unable to assess true medication use from ODB prescription claims data alone, thus estimates of asthma medications use may not correspond to exact doses taken by the individuals. Furthermore, our analyses on medication use were limited to children and young adults (≤25 years) who do not have a private plan and are therefore eligible for ODB. This may limit the generalizability of the findings to the general children and young adult population. As well, we were unable to assess changes, if any, to health-related quality of life for individuals managed largely by telemedicine compared to in-person care. On the other hand, the strength of population-based data from a single payer organized health system is that it allowed for complete capture and participant follow-up, a large sample size, and high power, which was necessary to study the trend of HSU from the pre-pandemic to current period (22 months during the pandemic and 14 months pre-pandemic). Asthma affected more than 700,000 children and young adults in Ontario, Canada. To our knowledge, this study is the first that quantified the impact of the pandemic in this population in Canada, and with the longest and most recent follow-up data.

## Conclusions

This Canadian population-based study demonstrated reduced ambulatory care, ED visits, hospitalization, ICS, and SABA prescription fills in children and young adults living with asthma during the COVID-19 pandemic. Our findings suggest that viral trigger avoidance strategies implemented during the pandemic may have contributed to the improved outcomes. Future research with longitudinal follow-up is needed to monitor the long-term impacts of the pandemic on health outcomes and quality of life in children and young adults with asthma.

## Supporting information

**S1 File.**
(DOCX)

**S1 Fig. Monthly asthma hospital admission rates by sex, income quintiles, and rural/urban residence, January 2019-December 2021.**
(TIF)

**S2 Fig. Monthly asthma emergency department (ED) visit rates by sex, income quintiles and rural/urban residence, January 2019-December 2021.**
(TIF)

**S3 Fig. Monthly physician office visit rates (in-person and virtual combined) by sex, income quintiles, and rural/urban residence, January 2019-December 2021.**
(TIF)

**S4 Fig. Monthly ICS and SABA prescription rates by sex, income quintiles, and rural/ urban residence, April 2019-December 2021.**
(TIF)

## Acknowledgments

This study was also supported by ICES (formerly the Institute for Clinical Evaluative Sciences). Parts of this material are based on data and information compiled and provided by the Canadian Institute for Health Information, Ontario Registrar General (ORG) information on deaths, the original source of which is ServiceOntario, and Cancer Care Ontario. Additionally, we thank IQVIA Solutions Canada Inc. for use of their Drug Information File. The analyses, conclusions, opinions, and statements expressed herein are those of the authors, and not necessarily those of the data sources; no endorsement is intended or should be inferred.

## Author Contributions

**Conceptualization:** Teresa To, Kimball Zhang.

**Data curation:** Kimball Zhang, Emilie Terebessy.

**Formal analysis:** Kimball Zhang, Jingqin Zhu.

**Funding acquisition:** Teresa To.

**Investigation:** Kimball Zhang.

**Methodology:** Kimball Zhang, Jingqin Zhu.

**Project administration:** Teresa To, Kimball Zhang, Emilie Terebessy.

**Supervision:** Teresa To, Jingqin Zhu.

**Validation:** Jingqin Zhu.

**Visualization:** Kimball Zhang, Emilie Terebessy.

**Writing – original draft:** Teresa To, Kimball Zhang.

**Writing – review & editing:** Kimball Zhang, Emilie Terebessy, Jingqin Zhu, Christopher Licskai.

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
