## [Decision Letter · Decision Letter 0]

11 Oct 2022

PONE-D-22-16786Healthcare Utilization in Canadian Children and Young Adults with Asthma during the COVID-19 PandemicPLOS ONE

Dear Dr. To,

Thank you for submitting your manuscript to PLOS ONE. After careful consideration, we feel that it has merit but does not fully meet PLOS ONE’s publication criteria as it currently stands. Therefore, we invite you to submit a revised version of the manuscript that addresses the points raised during the review process.

Please find the reviewers comments here below.==============================

We look forward to receiving your revised manuscript.

Kind regards,

Inge Roggen, M.D., Ph.D.

Academic Editor

PLOS ONE

“This study was funded by the Ontario Ministry of Health (MOH). Dr. Teresa To is funded by a Canadian Institutes of Health Research Tier 1 Canada Research Chair in Asthma. This study was also supported by ICES (formerly the Institute for Clinical Evaluative Sciences), which is funded by an annual grant from the Ontario Ministry of Health (MOH) and the Ministry of Long-Term Care (MLTC). Parts of this material are based on data and information compiled and provided by the Canadian Institute for Health Information, Ontario Registrar General (ORG) information on deaths, the original source of which is ServiceOntario, and Cancer Care Ontario. Additionally, we thank IQVIA Solutions Canada Inc. for use of their Drug Information File. The analyses, conclusions, opinions, and statements expressed herein are those of the authors, and not necessarily those of the funding or data sources; no endorsement is intended or should be inferred. The funders had no role in study design, in the collection, analysis, and interpretation of data, in the writing of the manuscript, and in the decision to submit the paper for publication.”

“Dr. TT receives a grant (HLTC3968IT) from the Ontario Ministry of Long-Term Care (https://www.ontario.ca/page/ministry-long-term-care). The funders had no role in study design, data collection and analysis, decision to publish, or preparation of the manuscript.”

6. We note that you have indicated that data from this study are available upon request. PLOS only allows data to be available upon request if there are legal or ethical restrictions on sharing data publicly. For more information on unacceptable data access restrictions, please see http://journals.plos.org/plosone/s/data-availability#loc-unacceptable-data-access-restrictions.

7. PLOS requires an ORCID iD for the corresponding author in Editorial Manager on papers submitted after December 6th, 2016. Please ensure that you have an ORCID iD and that it is validated in Editorial Manager. To do this, go to ‘Update my Information’ (in the upper left-hand corner of the main menu), and click on the Fetch/Validate link next to the ORCID field. This will take you to the ORCID site and allow you to create a new iD or authenticate a pre-existing iD in Editorial Manager. Please see the following video for instructions on linking an ORCID iD to your Editorial Manager account: https://www.youtube.com/watch?v=_xcclfuvtxQ.

Reviewers' comments:

Reviewer's Responses to Questions

**Comments to the Author**

1. Is the manuscript technically sound, and do the data support the conclusions?

Reviewer #1: Yes

Reviewer #2: Yes

2. Has the statistical analysis been performed appropriately and rigorously? 

Reviewer #1: Yes

Reviewer #2: Yes

3. Have the authors made all data underlying the findings in their manuscript fully available?

Reviewer #1: Yes

Reviewer #2: No

4. Is the manuscript presented in an intelligible fashion and written in standard English?

Reviewer #1: Yes

Reviewer #2: Yes

5. Review Comments to the Author

Reviewer #1: Well written paper. No incredible findings but the conclusion doesn't attempt to overstate the possible causes. Overall I have no concerns about the papers research methodology or findings. No spelling mistakes so that is great.

Reviewer #2: The authors use the national health administrative database for collecting data on healthcare use: disease burden (number of hospitalizations, ED, outpatient visits) and treatment prescription in the asthmatic population between 0-25 years.

The use of this database gives the opportunity to study a large sample size and to provide longitudinal data as highlighted in the discussion of this manuscript.

However, some concerns remain unresolved:

1/ The diagnosis of asthma is defined by the authors based on a low threshold (>or=1 hospitalization for asthma or >or=2 outpatient visits for asthma in 2 consecutive years). Especially in the lowest age category this could include an important selection bias. Children from 0-4y can have sporadic viral induced wheezing without underlying asthma as for example during RSV infection, necessitating hospitalization. This concern (of mixing ‘sporadic viral induced wheezing’ with asthma) has not been discussed in the manuscript and should at least earn some clarification in the discussion. An important amount of children who are in the age category 0-4 years prior to the pandemic are expected to outgrow this problem 2 years later. Besides, including the youngest age category increases the average monthly rate of all the outcomes measured, illustrated in Figure 1.

Subanalysis excluding this age-population would add valuable information.

2/ Study population is defined as children and young adults. In the introduction the authors describe that the target population is children and youth.

Youth is defined by WHO by individuals with age between 15 and 24 years old.

Young adults are defined between 12y and 30 years.

Why do authors choose the population until 25 years included and not until 24years? Has this been arbitrary chosen? Please clarify and adapt in the analysis if feasible.

3/ Rephrase the sentence, which sounds odd: P4 prescriptions on asthma medications filled and covered by OHIP were captured through the ODB database were identified…

4/ In the discussion part P18: rate of ICS increased early in the pandemic, but also SABA

6. PLOS authors have the option to publish the peer review history of their article (what does this mean?). If published, this will include your full peer review and any attached files.

Reviewer #1: No

Reviewer #2: No

---

## [Author Response · Author response to Decision Letter 0]

2 Nov 2022

All reviewer and editor comments have been addressed. Please see uploaded "Response to Reviewers" as well as revised "Cover Letter" and Data Availability section for this information.

---

## [Editor Report · Decision Letter 1]

28 Dec 2022

Healthcare Utilization in Canadian Children and Young Adults with Asthma during the COVID-19 Pandemic

PONE-D-22-16786R1

Dear Dr. To,

We’re pleased to inform you that your manuscript has been judged scientifically suitable for publication and will be formally accepted for publication once it meets all outstanding technical requirements.

Kind regards,

Inge Roggen, M.D., Ph.D.

Academic Editor

PLOS ONE
---

## [Editor Report · Acceptance letter]

4 Jan 2023

PONE-D-22-16786R1 

Healthcare utilization in Canadian children and young adults with asthma during the COVID-19 pandemic 

Dear Dr. To:

I'm pleased to inform you that your manuscript has been deemed suitable for publication in PLOS ONE. Congratulations! Your manuscript is now with our production department. 

Kind regards, 

on behalf of

Dr. Inge Roggen 

Academic Editor

PLOS ONE